# Proper Deficit Nitrogen Application and Irrigation of Tomato Can Obtain a Higher Fruit Quality and Improve Cultivation Profit

**Mengying Fan [1], Yonghui Qin [2], Xuelian Jiang [3],\*, Ningbo Cui [1],\*, Yaosheng Wang [4], Yixuan Zhang [1], Lu Zhao [1] and Shouzheng Jiang [1]**

[1]  State Key Laboratory of Hydraulics and Mountain River Engineering & College of Water Resource and Hydropower, Sichuan University, Chengdu 610065, China
[2]  Heng Yuan Survey and Design of Shandong Co., Ltd, Weifang 261000, China
[3]  Key Laboratory of Biochemistry and Molecular Biology in Universities of Shandong, Weifang University, Weifang 261000, China
[4]  Institute of Environment and Sustainable Development in Agriculture, Chinese Academy of Agriculture Science, Beijing 100081, China
\*  Correspondence: jiangxuelian1987@126.com (X.J.); cuiningbo@163.com (N.C.); Tel.: +86-150-0827-5024 (N.C.)

**Abstract:** Faced with severe global shortage of water and soil resources, studies on the integrated effect of water and nitrogen on tomato cultivation are urgently needed for sustainable agriculture. Two successive greenhouse experiments with three irrigation regimes (1, 2/3, 1/3 full irrigation) and four nitrogen levels (1, 2/3, 1/3, 0 nitrogen) were conducted; plant growth, fruit yield and quality were surveyed; and comprehensive quality and net profit were evaluated. The results show that water and nitrogen deficit decreased plant growth, evapotranspiration and yield while increasing production efficiency and fruit comprehensive quality. An antagonism effect from water and nitrogen application was found in tomato yield, organic acid, solids acid ratio, vitamin C and lycopene, whereas synergistic impact was observed in total soluble solids content. Water deficit had more significant effect on tomato yield and fruit quality parameters compared with that of nitrogen deficiency. Synthesizing the perspectives of yield, quality, resource productivity, market price index and profits, 1/3 full irrigation and 2/3 full nitrogen was the best strategy and could be recommended to farmers as an effective guidance for tomato production.

**Keywords:** deficit irrigation; nitrogen application; tomato; comprehensive quality; economic evaluation





## 1. Introduction

Tomato (*solanum lycopersicum* L.), as the most widely cultivated and globally popular vegetable, relies on its savory flavor and rich nutrition [1,2], and higher tomato consumption demonstrates antioxidant, anticancer, antimutagenic and antimicrobial effects on human health [3–5]. The global planting area of tomato reached $5.03 \times 10^6$ ha in 2019, with an annual production of $1.81 \times 10^8$ t [6]. As planting area and production of tomato are increasing, the fruit quantity can meet the requirements of the market, and consumers pay more attention to fruit quality [7]. Moreover, better fruit quality generally indicates higher economic benefits [8]. In addition, the reduction in agricultural irrigation and fertilization amount, improving water and fertilization productivity, is necessary for sustainable agricultural development in the context of severe water, soil and environmental resource scarcity [9].

Water and nitrogen application is vital to crop yield and quality [10–12]. Deficit irrigation decreased tomato evapotranspiration and yield [13–15] but improved fruit quality, including soluble solids, soluble sugar, vitamin C, polyphenols and lycopene content [16,17]. Gong [18] reported that 50% deficit irrigation decreased tomato evapotranspiration by 16–23%; Lu [19] reported regular deficit irrigation decreased tomato yield by 18.61 t ha$^{-1}$

on average, increased water use efficiency by 2.33 kg m$^{-3}$ and improved fruit quality. The decline in yield under deficit irrigation was mainly attributed to restrained photosynthesis due to water stress and impaired tomato physiological metabolites [20,21], whereas the increase in quality could be ascribed to the solute concentration caused by water loss in fruit [22]. A proper gradient of irrigation deficit could trade off high water use efficiency, fruit quality and acceptable yield reduction [23]. Plants can only absorb soluble nitrogen-containing ions such as $NO_3^-$ and $NH_4^+$ in soil through water migration [24]. Nitrogen is an irreplaceable composition of amino acids, proteins, nucleic acids and chlorophyll [25] that affects plant photosynthesis and metabolism directly [26,27] and further affects plant growth and fruitage [28]. Many studies have demonstrated that nitrogen application had positive effect on tomato yield and quality, including vitamin C, sugar–acid ratio, soluble sugar, total soluble solids and total phenols content [29–31]. However, there were other reports showing that nitrogen application rates had no significant effect on processing tomato yield [32] but only increased the aboveground biomass [33]. In addition, it was noted that nitrogen application had a nonnegligible negative impact on increasing nitrate content in tomato fruit [34]. Although multiple studies have reported the effect of water or nitrogen deficit on yield and individual fruit quality parameters, the results vary with tomato breeds, soil textures, climate and agronomic schemas in different experiments [35], and studies on the effect mechanisms of water and nitrogen are still urgent needed.

The integration of water and nitrogen is universally considered to exist in plant growth and fruitage. Zhou [36] found that nitrogen application could partially alleviate the biological stress caused by water stress on tomato plants and enhanced leaf water use efficiency. Nevertheless, irrigation can offset the negative effects of deficit nitrogen on crop productivity [37]. Appropriate sensor-based irrigation and nitrogen sustained high yield and reduced nitrogen leaching in low-holding-capacity soils [38,39], while an inappropriate water–nitrogen deficit level led to a decline in tomato yield and quality simultaneously [40]. Although the effect of the coupling of water and nitrogen is admittedly recognized on plant growth, evidence is still lacking to clarify their inner relationship. Some studies found no significant relationship between water and nitrogen's effects on yield in maize [41], watermelon [42] and tomato [34]. The inner relationship between water and nitrogen and their response threshold, which are indispensable to adjustable irrigation–nitrogen decision making, still remain elusive. Therefore, the marginal productivity efficiency and the comprehensive benefits based on yield, fruit quality, source efficiency and market price were calculated to evaluate the outcome of each application strategy. The aims of the present study are to: (1) investigate the effect of irrigation and nitrogen on tomato growth, yield, quality and water–nitrogen use efficiency; (2) clarify whether there is a synergistic or antagonistic relationship between water and nitrogen's effects on yield and different quality traits; (3) determine the preferable water and nitrogen application strategy based on comprehensive benefit analysis and provide a direct scientific guidance for local tomato cultivation industry.

## 2. Materials and Methods

### 2.1. Experimental Site

The greenhouse experiments were conducted from 13 April to 6 July (first season) and 13 August to 28 November (second season) in 2019 at a commercial company located in Gaomi City (latitude 36°38′ N, longitude 112°56′ E, altitude 26.03 m), Shandong Province, in Northern China. The site is located in a monsoon climate with annual precipitation of 646 mm, pan evaporation of 1838 mm, temperature of 11.7 °C, and duration of mean annual sunshine over 2800 h. The greenhouse is 80 m in length and 12 m in width, covering an area of 960 m$^2$. The experimental soil is clay loam with the average dry bulk density of 1.38 g·cm$^{-3}$. The total available N, P and K content for 0–1.0 m soil depth initially are 84.3 mg·kg$^{-1}$, 102 mg·kg$^{-1}$ and 130 mg·kg$^{-1}$, respectively. The field water capacity ($\theta_{FC}$) and the wilting coefficient for 0–0.8 m depth determined by soil water absorption experiment are 0.33 cm$^3$·cm$^{-3}$ and 0.14 cm$^3$·cm$^{-3}$, respectively.

## 2.2. Experimental Design

The experimental tomato (*Solanum lycopersicum* L.) variety was Baoli 3 in both seasons. The tomato plants with similar heights were transplanted to the plots on 13 April in the first season and 13 August in the second season. Three growth stages, i.e., seedling stage (from transplant to first fruit set), flowering and fruit development stage (from first fruit set to first fruit maturity) and fruit maturation and harvest stage (from first fruit maturity to uprooting crops after all fruit is harvested), were divided according to local observations. The first season ended on 26 July and the second on 28 November.

Three irrigation levels consisted of full irrigation (W1), 2/3 full irrigation (W2/3) and 1/3 full irrigation (W1/3). Four urea nitrogen levels included full nitrogen (N1), 2/3 full nitrogen (N2/3), 1/3 full nitrogen (N1/3) and no nitrogen (N0) at the whole growth stage. This yielded twelve treatments in a completely randomized block design, and each treatment was replicated thrice. The size of each plot was 2.8 m × 6 m, and the plots were separated by 1 m deep and 2 mm thick acrylic flap. W1 was irrigated to $90 \pm 3\%$ of $\theta_{FC}$ when the soil water content within 0.6 m at the seedling stage and 0.8 m in other stages decreased to $75 \pm 3\%$ of $\theta_{FC}$. The nitrogen amount of N1 treatment was consistent with that of local management. The irrigation treatment was irrigated at rate of water requirements based on control treatment W1N1, and the nitrogen treatment was fertilized at rate of N use in W1N1. Furthermore, concerning the relatively low available N content in soil and severe deficit in first season, the same amount of nitrogen application was implemented in second season. The irrigation and fertilization time for deficit treatments was the same as that of W1N1. The description of irrigation and fertilization amount under different treatments is shown in Table 1. The irrigation pattern was dripping irrigation, and each plot had an individual 6 m length branch. Twelve water drippers with a flow of 1.6 L/hour were evenly distributed in the branch, each branch pipe was separately installed with a water meter, and a Venturi fertilizing tank was used to record the irrigation and fertilization amount. Three tomato plants were planted in a plot and uniformly pruned (removing the secondary shoots and only leaving the main stem) in flowering and fruit development stage according to the growth condition.

**Table 1.** Description of irrigation and nitrogen amount in different treatments.

| Num | Treatment | Description (at Whole Season) | First Season | | Second Season | |
|---|---|---|---|---|---|---|
| | | | I | N | I | N |
| T1 | $W_1N_1$/CK | full irrigation and full nitrogen | 260.4 (10) | 14.7 (3) | 184.3 (10) | 19.6 (4) |
| T2 | $W_1N_{2/3}$ | full irrigation and 2/3 full nitrogen | 260.4 (10) | 9.8 (3) | 184.3 (10) | 15.8 (4) |
| T3 | $W_1N_{1/3}$ | full irrigation and 1/3 full nitrogen | 260.4 (10) | 4.9 (3) | 184.3 (10) | 12.0 (4) |
| T4 | $W_1N_0$ | full irrigation and no nitrogen | 260.4 (10) | 0.0 (0) | 184.3 (10) | 8.2 (1) |
| T5 | $W_{2/3}N_1$ | 2/3 full irrigation and full nitrogen | 180.6 (10) | 14.7 (3) | 139.4 (10) | 19.6 (4) |
| T6 | $W_{2/3}N_{2/3}$ | 2/3 full irrigation and 2/3 full nitrogen | 180.6 (10) | 9.8 (3) | 139.4 (10) | 15.8 (4) |
| T7 | $W_{1/3}N_{1/3}$ | 2/3 full irrigation and 1/3 full nitrogen | 180.6 (10) | 4.9 (3) | 139.4 (10) | 12.0 (4) |
| T8 | $W_1N_0$ | 2/3 full irrigation and no nitrogen | 180.6 (10) | 0.0 (0) | 139.4 (10) | 8.2 (1) |
| T9 | $W_{1/3}N_1$ | 1/3 full irrigation and full nitrogen | 104.9 (10) | 14.7 (3) | 87.5 (10) | 19.6 (4) |
| T10 | $W_{1/3}N_{2/3}$ | 1/3 full irrigation and 2/3 full nitrogen | 104.9 (10) | 9.8 (3) | 87.5 (10) | 15.8 (4) |
| T11 | $W_{1/3}N_{1/3}$ | 1/3 full irrigation and 1/3 full nitrogen | 104.9 (10) | 4.9 (3) | 87.5 (10) | 12.0 (4) |
| T12 | $W_{1/3}N_0$ | 1/3 full irrigation and no nitrogen | 104.9 (10) | 0.0 (0) | 87.5 (10) | 8.2 (1) |
| | stage I | seedling stage | 13 Apr.–7 May | | 13 Aug.–6 Sep. | |
| | stage II | flowering and fruit development stage | 8 May–20 Jun. | | 7 Sep.–13 Oct. | |
| | stage III | fruit maturation and harvest stage | 21 Jun.–26 Jul. | | 14 Oct.–28 Nov. | |

Notes: CK means control treatment; T1 indicates the first treatment and so on for the other treatments; the number in brackets represents the irrigation and fertilization times; I means irrigation amount (mm); N means nitrogen application (g·m$^{-2}$).

*2.3. Measurements*

2.3.1. Meteorological Variables

The solar radiation (Ra), air temperature (Ta) and relative humidity (RH) in the two seasons were recorded continuously using a standard automatic weather station (Hobo, Onset Computer Crop, Bourne, MA, USA) installed at the center of the greenhouse. All meteorological mean variables every 30 min were calculated automatically by a data logger, and the daily average value of Ra, Ta and RH are shown in Figure 1.

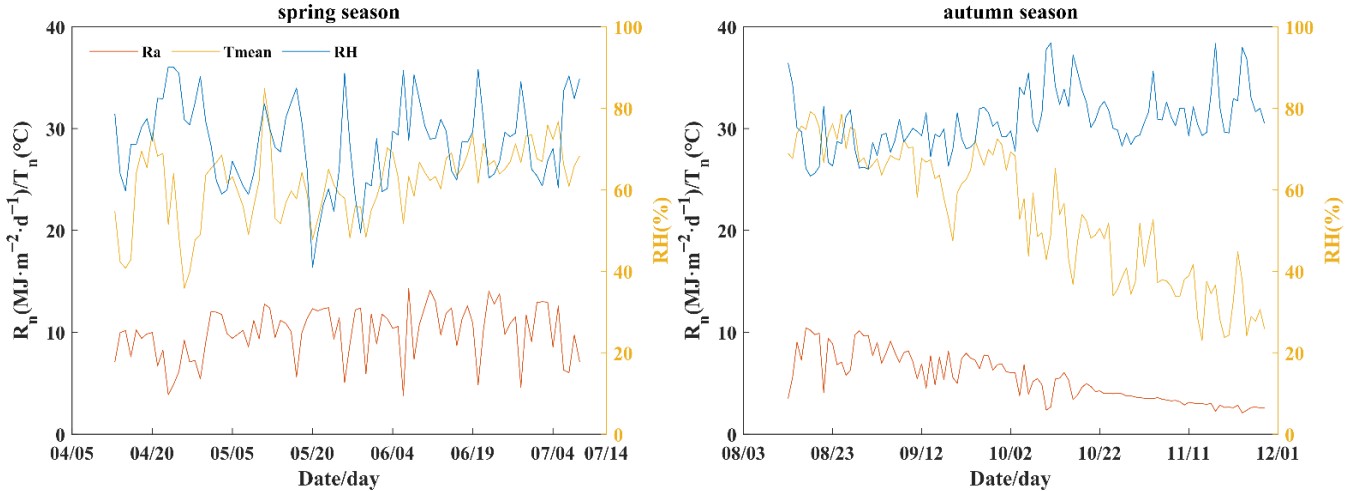

**Figure 1.** The basic meteorological information during the growth stages.

2.3.2. Evapotranspiration

Crop evapotranspiration (ET) was estimated by soil water balance method [43] as follows:

$$ET = P + I + W - R - D - \Delta W, \tag{1}$$

where P is precipitation (mm); I is irrigation amount (mm); W is capillary rise to the root zone (mm); R is surface runoff (mm); D is drainage from the root zone (mm); and $\Delta W$ is the change in soil water content (mm). $\Delta W$ was calculated as follows:

$$\Delta W = H(\theta_i - \theta_{i-1}), \tag{2}$$

where H is the depth of plant root zone (m); $\theta_i$ and $\theta_{i-1}$ are the mean water contents in the root zone at time i and i − 1, respectively.

Since there is no precipitation in greenhouse, P and R can be negligible. The groundwater level was lower than 15 m below the ground surface according to the local observation, so W was also negligible. D can be ignored because the irrigation amount was always within the field water capacity. Thus, Equation 1 is simplified as:

$$ET = I - H(\theta_i - \theta_{i-1}), \tag{3}$$

2.3.3. Plant Growth

Plant height, stem diameter and leaf index (LAI) were measured at intervals of 7–10 days during the whole growth period. Leaf length and the maximum width were measured, and the leaf area was determined by the sum of the rectangular area of each completely developed leaf (the product of leaf length and maximum width) multiplied by a parameter of 0.64 [36,44]. The LAI was the ratio of leaf area to land area of each plant. Chlorophyll content was measured by a handled chlorophyll analyzer (SPAD502, Spectrum, Aurora, IL, USA, 0.1) every 7–10 days.

### 2.3.4. Yield, Water and Nitrogen Use Efficiency

Fifteen plants in each plot were randomly selected for measuring the yield, and fruit weight after maturity was measured by an electronic scale. The total yield (Y, t·ha$^{-1}$) and mean single fruit weight was then calculated. Water use efficiency (WUE, kg·m$^{-3}$), irrigation water use efficiency (WUEI, kg·m$^{-3}$) and application nitrogen use efficiency (NUE, kg·g$^{-1}$) were calculated as follows:

$$WUE = Y/ET \times 100, \tag{4}$$

$$WUEI = Y/I \times 100, \tag{5}$$

$$NUE = Y/N \times 0.1, \tag{6}$$

where Y is the yield (t·ha$^{-1}$); I is the irrigation quantity (mm); and N is the nitrogen application amount (g·m$^{-2}$).

### 2.3.5. Fruit Quality

Fruit quality parameters were measured at fruit maturation and harvest stage. Total soluble solids content (TSS) was measured by a handheld refractometer (PAL-BX/ACID 3, ATAGO, Tokyo, Japan, 0.1 Brix). Organic acid (OA) was titrated with 0.1 mol·L$^{-1}$ NaOH solution and the solids-acid ratio (SAR) was defined as the ratio of TSS to OA. Vitamin C (VC) content was measured using 2,6-dichloroindophenol titrimetric method (A009-1-1, Nanjing Jiancheng bioengineering institute, China, 0.1 ug·ml$^{-1}$) [45]. Lycopene content (Lyc) was measured by spectrophotometric method (FT-P6141Z, Fantaibio, China, 0.1 ug·mL$^{-1}$) [46,47]. Fruit firmness was measured by a hardness tester (GY-4, Handpi, Zhejiang, China, 0.01 kg·cm$^{-2}$).

### *2.4. The Calculation of Comprehensive Quality*

Tomato quality is an overall result of individual parameters, and the responses of individual fruit quality parameters to irrigation and nitrogen treatments are different, which affects the determination of the treatment that has the best fruit quality. Thus, the comprehensive fruit quality was evaluated using the technique for order preference by similarity to ideal solution (TOPSIS), combined with analytic hierarchy process (AHP), which is briefly outlined below.

(1)　Normalize individual fruit quality parameters. The low optimal parameter is converted into high optimal parameter as follows:

$$x_{ij} = \frac{1}{x_{ij}^*}, \quad i = 1, 2 \cdots m; \ j = 1, 2 \cdots n, \tag{7}$$

where $x_{ij}$ is the forward original quality value of i-th treatment and j-th fruit quality parameter. In this study, m = 12 and n = 7; $x_{ij}^*$ is the antidromic original quality value; and only OA was considered to be small optimal index in this study.

Then, $x_{ij}$ is normalized as follows:

$$z_{ij} = \frac{\left| x_{ij} - x_{bestj} \right|}{\sqrt{\sum_{i=1}^{n} \left( x_{ij} - x_{bestj} \right)^2}}, \quad i = 1, 2 \cdots m; \ j = 1, 2 \cdots n, \tag{8}$$

where $z_{ij}$ is the positively standardized quality value of i-th treatment and j-th fruit quality parameter; $x_{bestj}$ is the best value of j-th parameter among all treatments.

(2)　Define the best and worst ideal solutions:

$$Z^+ = \left( Z_1^+, Z_1^+, \ldots Z_j^+, \ldots Z_m^+ \right), \ Z_j^+ = \max \left\{ z_{1j}, z_{2j}, \ldots, z_{nj} \right\}, \tag{9}$$

$$Z^- = \left(Z_1^-, Z_1^-, \ldots Z_j^-, \ldots Z_m^-\right), \ Z_j^- = \min\{z_{1j}, z_{2j}, \ldots, z_{nj}\}, \tag{10}$$

where $Z^+$ is the defining maximum matrix; $Z_j^+$ is the maximum value of parameter j; $Z^-$ is the defining minimum matrix; and $Z_j^-$ is the minimum value of parameter j.

(3)　Calculate the distance using AHP weights:

$$D_i^+ = \sqrt{\sum_{j=1}^m \omega_j \left(Z_j^+ - z_{ij}\right)^2}, \ D_i^- = \sqrt{\sum_{j=1}^m \omega_j \left(Z_j^- - z_{ij}\right)^2}, \tag{11}$$

where $D_i^+$ is the distance between i treatment and the maximum value; $D_i^-$ is the distance between i treatment and the minimum value; $\omega_j$ is the weight of index j determined by AHP method [48].

(4)　Compute the comprehensive index under different treatments (Qi):

$$Q_i = \frac{D_i^-}{D_i^+ + D_i^-}, \tag{12}$$

*2.5. Economic Analysis*

The economic benefits are related to both yield and fruit quality, and better fruit quality usually indicates higher sale prices. Thus, an economic profit analysis considering comprehensive quality and yield is necessary to determine the optimal treatment.

Relative sale price considering the comprehensive quality was calculated as:

$$p_i = p_c \left(1 + R\left(\frac{Q_i}{Q_{ck}} - 1\right)\right), \tag{13}$$

where $p_i$ is the price of different fruit quality ($\cdot kg^{-1}$); $p_c$ is the sale price of CK treatment ($\cdot kg^{-1}$), which was defined as 1.5 according to the market sale price in 2019; R is the price index, representing the fluctuation of tomato prices with quality; and $Q_i$ and $Q_{ck}$ are the comprehensive quality index under different irrigation and nitrogen treatment and CK treatment, respectively.

Total cost under different treatment was determined by:

$$C_c = I \times c_w + N \times c_f + C_s, \tag{14}$$

where $C_c$ is the total cost ($); I is the irrigation amount ($m^3 \cdot ha^{-1}$); $c_w$ is the unit price of agricultural water ($\cdot m^{-3}$), which is defined as 0.15 according to the water conservancy company sale price in 2019; N is nitrogen amount applied ($kg \cdot ha^{-1}$); $c_f$ is the unit price of nitrogen fertilizer ($\cdot kg^{-1}$), defined as 2.7 according to the market sale price in 2019; and $C_s$ is the fixed cost under different treatments ($), which does not change due to different treatments, including the cost of greenhouse drip irrigation project, land rent, other facilities costs, $282,750 one-time investment expense for 20 years usage expectation per ha, labor costs (land leveling, tomato interruption, weeding, fertilization, spraying, harvesting and packaging) totaling $652 per ha per year and other consumables costs (fertilizers other than nitrogen, herbicide and insecticide) totaling $580 per ha per year.

Net profit was determined as:

$$C_n = Y_i p_i - C_c, \tag{15}$$

where $C_n$ is the net profit per hectare ($); $Y_i$ is the yield in different treatments ($kg \cdot ha^{-1}$).

The profit change was calculated as:

$$\Delta P_\% = \frac{C_{ni} - C_{nck}}{C_{nck}} \times 100\%, \tag{16}$$

where $\Delta P_\%$ is the change in net profit, %; $C_{ni}$ is the net profit of i treatment; and $C_{nck}$ is the net profit of CK treatment.

### 2.6. Statistical Analysis

The differences among the treatments were compared by two-way ANOVA and Duncan's multiple range test using SPSS 26.0 (IBM statistics, Armonk, NY, USA). The correlation between independent and dependent variables was determined by the Pearson correlation analysis.

## 3. Results

### 3.1. Effect of Deficit Irrigation and Nitrogen on Plant Growth

The variation in plant height, stem diameter, leaf area index and chlorophyll content of tomato under different irrigation and nitrogen treatments in the second season is shown in Figure 2. The plant height increased rapidly in the first two stages and remained stable after reaching its maximum at the beginning of stage III. The stem diameter showed similar change tendency to that of plant height, except for reaching its maximum at the beginning of stage II. Water deficit had no obvious effect on plant height but significantly reduced stem diameter (Figure 2a–f) due to the infinite-growth tomato variety, which may not be very sensitive to water stress in height growth. Nitrogen application improved plant height and stem diameter in low-water treatments (Figure 2c,f), indicating that nitrogen could alleviate the inhibition caused by water deficit as a vital nutritious element of plant growth. Compared with CK, plant height and stem diameter of T12 in stable point (November 10) decreased by 5.88% and 7.62%, respectively.

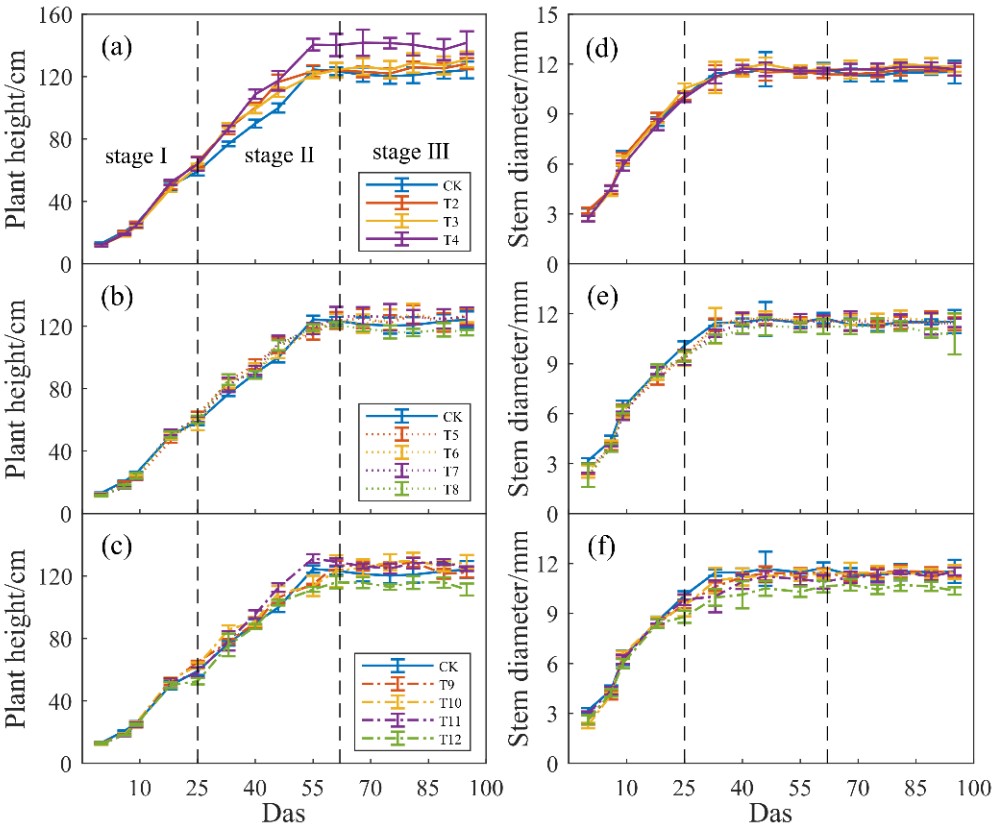

**Figure 2.** *Cont.*

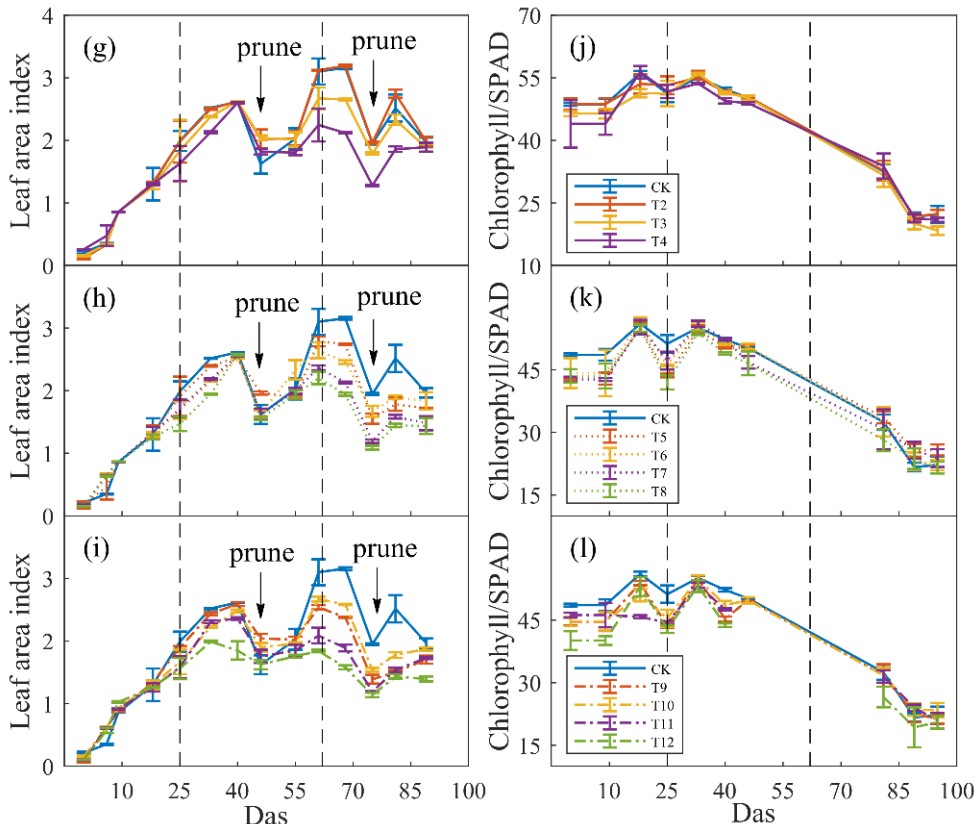

**Figure 2.** The plant height (h), stem diameter (D), leaf area index (LAI) and chlorophyll content (SPAD) under different water and nitrogen treatments in the second season of 2019. Notes: (**a,d,g,j**) show sufficient irrigation groups of CK, T2, T3 and T4, respectively; (**b,e,h,k**) represent moderate water deficit groups of T5, T6, T7 and T8, respectively; (**c,f,i,l**) indicate severe water deficit groups of T9, T10, T11 and T12, respectively.

Leaf area index (LAI) and chlorophyll content increased at stage I and II and then decreased due to leaf wilting and yellowing (Figure 2g–l). The significant drops on 27 September and 27 October were caused by pruning. Both water and nitrogen deficit reduced LAI observably. Compared with CK, LAI decreased from 26.6% (W2/3) to 33.2% (W1/3) for water deficit treatments and 13.1% (N2/3) to 29.1% (N0) for nitrogen deficit treatments (in maximum point, 20 October). Chlorophyll content decreased by 8.4% (N0) and 9.6% (W1/3) on August 22 at stage I but had no obvious variance after stage I (on 15 September and 10 November, $p > 0.05$).

### 3.2. Effect of Deficit Irrigation and Nitrogen on Evapotranspiration, Yield, Water and Nitrogen Use Efficiency

The evapotranspiration (ET) at the whole growth stage varied from 300.99 to 173.77 mm in the first season and from 239.80 to 161.49 mm in the second season (Table 2). Both irrigation and nitrogen fertilization decreased ET significantly (Table 2, $p < 0.01$), and the decline in ET reached 38.8% (T9) and 5.7% (T4) in the first season and 23.6% (T9) and 11.6% (T4) in the second season.

The effect of water and nitrogen on tomato yield was significant (Table 2, $p < 0.05$). Compared with CK, yield decreased by 39.1% (T9) and 11.0% (T4) in the first season and 10.1% (T9) and 4.9% (T4) in the second season.

Compared with CK, water use efficiency (WUE) increased first and then decreased with the decline in water supply in the first season but had no significant differences in the second season (Table 2). The decrease in water use efficiency in W1/3 was caused by sharp yield recession, and there was no significant effect of nitrogen on WUE.

**Table 2.** Evapotranspiration (ET), yield and product efficiency (WUE, WUEI and NUE) under different water and nitrogen treatments in the first and second seasons of 2019.

| Treatment | | ET (mm) | Yield (t·ha$^{-1}$) | WUE (kg·m$^{-3}$) | WUEI (kg·m$^{-3}$) | NUE (kg·g$^{-1}$) |
|---|---|---|---|---|---|---|
| | CK | 299.41 a | 66.09 a | 22.07 ab | 25.38 cd | 0.45 d |
| | T2 | 300.99 a | 66.46 a | 22.08 ab | 25.52 cd | 0.68 c |
| | T3 | 288.44 ab | 65.33 a | 22.65 ab | 25.09 cd | 1.33 a |
| | T4 | 282.33 b | 58.79 ab | 20.82 b | 22.58 d | |
| | T5 | 245.79 c | 68.28 a | 27.78 a | 37.81 a | 0.46 d |
| First season | T6 | 243.26 c | 63.84 a | 26.24 ab | 35.35 ab | 0.65 c |
| | T7 | 239.89 c | 48.23 bc | 20.11 b | 26.71 bcd | 0.98 b |
| | T8 | 225.05 d | 47.98 bc | 21.32 ab | 26.57 bcd | |
| | T9 | 204.59 e | 40.23 c | 19.66 b | 38.35 a | 0.27 e |
| | T10 | 190.83 f | 38.48 c | 20.16 b | 36.68 a | 0.39 de |
| | T11 | 181.28 fg | 35.08 c | 19.35 b | 33.44 abc | 0.72 c |
| | T12 | 173.77 g | 34.32 c | 19.75 b | 32.72 abc | |
| | W | 0.000 ** | 0.000 ** | 0.022 * | 0.000 ** | 0.000 ** |
| Sig test | N | 0.000 ** | 0.019 * | 0.278 ns | 0.017 * | 0.011 * |
| | W*N | 0.326 ns | 0.482 ns | 0.397 ns | 0.608 ns | 0.135 ns |
| | CK | 239.80 a | 59.92 a | 24.99 ab | 32.51 cd | 0.29 e |
| | T2 | 229.59 ab | 57.55 ab | 25.07 ab | 31.23 cd | 0.34 de |
| | T3 | 225.45 b | 54.32 ab | 24.09 ab | 29.47 d | 0.42 bcd |
| | T4 | 212.01 c | 56.97 ab | 26.87 ab | 30.91 cd | 0.61 a |
| | T5 | 206.46 c | 49.36 abc | 23.91 ab | 35.41 bcd | 0.25 e |
| Second season | T6 | 206.69 c | 56.86 ab | 27.51 ab | 40.79 bcd | 0.36 cde |
| | T7 | 205.51 c | 51.77 abc | 25.19 ab | 37.14 bcd | 0.43 bcd |
| | T8 | 183.56 d | 42.14 bc | 22.96 b | 30.23 cd | 0.51 ab |
| | T9 | 183.16 d | 53.85 abc | 29.40 ab | 61.54 a | 0.28 e |
| | T10 | 177.75 d | 56.79 ab | 31.95 a | 64.90 a | 0.36 cde |
| | T11 | 176.77 d | 41.46 bc | 23.46 b | 47.39 b | 0.35 de |
| | T12 | 161.49 e | 38.00 c | 23.53 b | 43.43 bc | 0.46 bc |
| | W | 0.000 ** | 0.027 * | 0.392 ns | 0.000 ** | 0.134 ns |
| Sig test | N | 0.000 ** | 0.034 * | 0.221 ns | 0.000 ** | 0.000 ** |
| | W*N | 0.749 ns | 0.456 ns | 0.384 ns | 0.001 ** | 0.248 ns |

Notes: lowercase letters following the data indicate significant differences by Duncan's test at $p < 0.05$ level; * means statistically significant with $p < 0.05$; ** notes statistically extreme significance with $p < 0.01$; ns represents statistically insignificant with $p > 0.05$.

Irrigation water use efficiency (WUEI) significantly increased with the reduction in irrigation amount and deceased with the reduction in nitrogen amount in both seasons. Oppositely, nitrogen use efficiency (NUE) decreased with the reduction in irrigation amount and increased with the reduction in nitrogen amount (Table 2). WUEI was generally higher than WUE, since a plant can use the water in soil that was stored prior to development stage. The maximum of WUEI was 38.35 kg·m$^{-3}$, found in T9, in the first season and 64.90 kg·m$^{-3}$, observed in T10, in the second season. T3 in the first season had a much higher NUE of 1.33 kg·g$^{-1}$ than any other treatments. It was interesting that the two-way ANOVA results of water and nitrogen interaction on ET, yield and efficiency were not significant except for WUEI in the second season (Table 2).

### 3.3. Effect of Deficit Irrigation and Nitrogen on Fruit Quality

The TSS, SAR and VC significantly increased with the increase in irrigation deficit, while SW and OA decreased with the increase in irrigation deficit (Table 3). Compared with CK, T9 increased TSS by 22.9% and 37.0%, SAR by 79.3% and 51.6% and VC by 112.3% and 129.9% in the first and second season, respectively. SW decreased by 14.6% and 12.8%, and OA decreased by 23.8% and 11.1% in the first and second seasons, respectively. Lyc decreased with the decline in water in the second season, but there was no significant

variance in the first season. Fn of W1/3 increased 12.7% in the first season but did not change obviously in the second season.

**Table 3.** Single fruit quality parameters under different water and nitrogen treatments in the first and second seasons of 2019.

| Treatment | | SW (g) | TSS (°Brix) | OA (%) | SAR (ratio) | VC (mg·kg⁻¹) | Lyc (mg·kg⁻¹) | Fn (kg·cm⁻²) |
|---|---|---|---|---|---|---|---|---|
| First season | CK | 93.08 ab | 4.85 fg | 0.42 a | 10.54 d | 1.06 b | 26.40 a | 4.15 abc |
| | T2 | 95.86 a | 5.38 def | 0.38 ab | 14.01 cd | 1.44 ab | 24.00 a | 3.45 bc |
| | T3 | 95.94 a | 5.30 def | 0.32 bcd | 16.53 bcd | 1.39 ab | 22.03 a | 3.32 bc |
| | T4 | 88.25 abc | 4.38 g | 0.33 bcd | 12.93 cd | 0.92 b | 17.11 a | 3.19 bc |
| | T5 | 99.69 a | 5.06 ef | 0.34 abcd | 15.83 bcd | 1.29 b | 23.76 a | 3.12 bc |
| | T6 | 91.79 abc | 5.61 bcde | 0.36 abc | 15.34 bcd | 1.67 ab | 20.20 a | 3.73 abc |
| | T7 | 85.32 abc | 5.83 abcd | 0.30 bcde | 20.00 bc | 1.03 b | 18.46 a | 3.03 c |
| | T8 | 84.07 abc | 5.55 bcde | 0.26 de | 21.97 ab | 1.08 b | 20.70 a | 3.24 bc |
| | T9 | 79.50 bcd | 5.96 abc | 0.32 bcd | 18.90 bc | 2.25 a | 20.84 a | 3.93 abc |
| | T10 | 76.88 cd | 6.24 a | 0.27 cde | 22.59 ab | 2.20 a | 27.20 a | 5.24 a |
| | T11 | 69.33 d | 6.05 ab | 0.33 bcd | 18.35 bc | 1.54 ab | 17.01 a | 4.71 abc |
| | T12 | 67.00 d | 5.42 cde | 0.23 e | 28.63 a | 1.06 b | 13.63 a | 4.83 ab |
| Sig test | W | 0.000 ** | 0.000 ** | 0.001 ** | 0.000 ** | 0.012 * | 0.715 ns | 0.001 ** |
| | N | 0.002 ** | 0.000 ** | 0.003 ** | 0.027 * | 0.022 * | 0.210 ns | 0.709 ns |
| | W*N | 0.384 ns | 0.034 * | 0.254 ns | 0.108 ns | 0.321 ns | 0.773 ns | 0.589 ns |
| Second season | CK | 129.56 ab | 4.16 f | 0.36 bcd | 11.76 d | 1.34 g | 46.08 a | 3.18 a |
| | T2 | 132.44 ab | 5.35 abcd | 0.37 bc | 15.09 bc | 1.40 fg | 44.01 ab | 3.16 a |
| | T3 | 127.35 ab | 5.52 abc | 0.46 a | 12.65 d | 1.91 de | 40.83 abc | 3.76 a |
| | T4 | 122.71 ab | 5.00 de | 0.37 bc | 13.76 cd | 1.73 efg | 30.06 bc | 3.79 a |
| | T5 | 126.47 ab | 5.44 abcd | 0.48 a | 11.46 d | 2.23 bcd | 35.10 abc | 4.30 a |
| | T6 | 108.33 ab | 5.55 ab | 0.39 b | 15.62 abc | 2.49 b | 37.12 abc | 3.36 a |
| | T7 | 115.53 ab | 5.14 bcde | 0.34 bcde | 15.15 bc | 2.63 b | 35.67 abc | 3.26 a |
| | T8 | 150.01 a | 5.07 cde | 0.30 e | 16.46 ab | 1.80 ef | 31.39 abc | 3.54 a |
| | T9 | 113.02 ab | 5.70 a | 0.32 cde | 17.83 a | 3.08 a | 29.25 bc | 3.40 a |
| | T10 | 116.37 ab | 5.28 abcd | 0.31 de | 16.94 ab | 2.43 b | 30.84 bc | 3.45 a |
| | T11 | 93.50 b | 5.47 abcd | 0.29 e | 17.89 a | 2.37 bc | 29.97 bc | 2.83 a |
| | T12 | 84.60 b | 4.70 e | 0.30 e | 16.46 ab | 2.00 cde | 27.86 c | 3.48 a |
| Sig test | W | 0.011 * | 0.008 ** | 0.000 ** | 0.000 ** | 0.000 ** | 0.011 * | 0.879 ns |
| | N | 0.819 ns | 0.000 ** | 0.000 ** | 0.008 ** | 0.001 ** | 0.199 ns | 0.941 ns |
| | W*N | 0.466 ns | 0.000 ** | 0.000 ** | 0.003 ** | 0.000 ** | 0.755 ns | 0.962 ns |

Notes: SW, TSS, OA, SAR, VC, Lyc and Fn indicated tomato single weight, total soluble solids, organic acids, solid–acid content ratio, vitamin C content, lycopene content and fruit firmness, respectively. lowercase letters following the data indicate significant differences by Duncan's test at $p < 0.05$ level; * means statistically significant with $p < 0.05$; ** notes statistically extreme significance with $p < 0.01$; ns represents statistically insignificant with $p > 0.05$.

SW declined with the deficit of nitrogen in the first season, but there was no obvious variance in the second season. Compared with N1, nitrogen deficit significantly improved fruit TSS by 8.8% (N2/3 and N1/3), while it decreased fruit TSS by 3.0% under N0 in the first season, and fruit TSS increased by 7.7% with N2/3 and N1/3 while decreasing by 1.4% with N0 under nitrogen deficit in the second season. Similarly, average VC in two seasons of N2/3 increased by 9.7% and decreased by 17.9% under N0. However, nitrogen had no significant effect on Lyc and Fn.

The two-way ANOVA showed that the water more pronouncedly affected the results than the nitrogen treatments, indicating that, compared with nitrogen, water occupied a dominant position in tomato yield and fruit quality, in agreement with the experimental results in northwest China [35,49].

### 3.4. The Interactive Relationship between Water and Nitrogen Application

Since the two-way ANOVA results indicate that the interaction of water and nitrogen was weak and unclear, marginal values (slope) were calculated to further analyze their relationship, and the results are shown in Figure 3. Tomato marginal yield of W2/3 and W1/3 increased with nitrogen application, and the increasing weakened in W1, indicating that an antagonism effect between water and nitrogen was only found under W1~W2/3 and N1~N1/3 (Figure 3). Although both water and nitrogen promoted tomato yield, their effect intensity was gradually decreased with source supply saturation, which finally caused low efficiency [8]. For OA and SAR, a reciprocal effect from water with nitrogen occurred only under N2/3~N1/3. Water and nitrogen exhibited antagonism on SW, VC and Lyc under N1~N1/3 and synergy on TSS under N1~N1/3 and W1~W2/3. The interaction of water and nitrogen tended to disappear when one of them was extremely deficient, and water took a dominated position in the interaction [49].

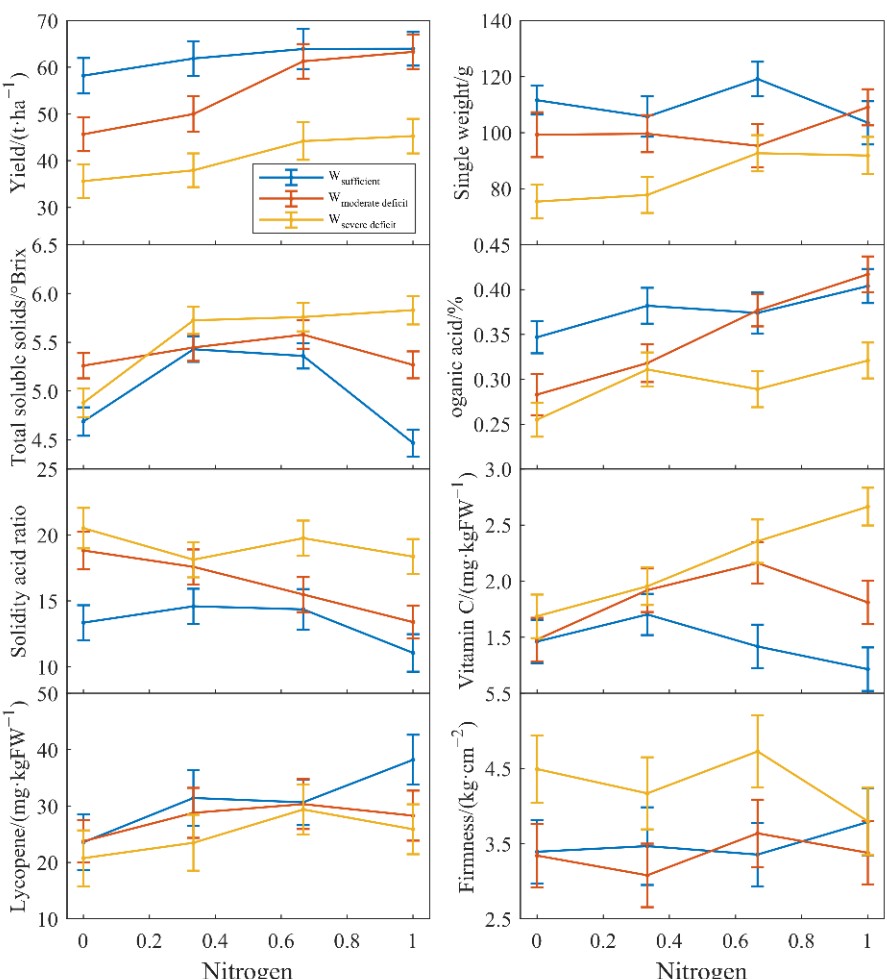

**Figure 3.** The interaction effect of water and nitrogen on yield, single fruit qualities parameters and comprehensive fruit quality in the first and second seasons of 2019. Notes: the vertical axis is the estimated marginal mean value of each parameter; the horizontal axis is the gradient of nitrogen including full nitrogen, mild stress, medium stress and heavy stress.

### 3.5. Comprehensive Quality Assessment and Economic Analysis

The weights of each individual quality index calculated by Analytic Hierarchy Process (AHP) are shown in Table 4. The random consistency ratio CR of the judgment matrix was 0.00025 < 0.1, indicating that the calculated weights met the requirements of pairwise comparison consistency. The comprehensive evaluation of the technique for order preference by similarity to ideal solution method (TOPSIS) showed that the individual fruit quality TSS

and VC were the main factors affecting overall score in both seasons (Table 4), while the effect of other parameters was restricted by small weights or slight fluctuation of different treatments. Due to the correlation between pivotal quality parameters and comprehensive quality, it was feasible to focus on TSS and VC to evaluate tomato fruit in the absence of quality tests. The assessing results of comprehensive quality in both seasons were roughly consistent. T10 reached the highest quality value (Q) of 0.770 and ranked no. 1 in the first season, and T9 reached Q of 0.599 and 0.641, ranking no. 2 and no. 1 in the first and second seasons, respectively. Treatments with severe water deficit and mild nitrogen stress (T9, T10) attained the best integrated quality.

**Table 4.** TOPSIS analysis of comprehensive fruit quality under different water and nitrogen treatments in the first and second seasons of 2019.

| Treatment Weight | | SW 0.110 | TSS 0.131 | OA 0.088 | SAR 0.180 | VC 0.173 | Lyc 0.198 | Fn 0.116 | $D^+$ - | $D^-$ - | Q - | Rank - |
|---|---|---|---|---|---|---|---|---|---|---|---|---|
| First season | CK | 0.312 | 0.255 | 0.210 | 0.164 | 0.207 | 0.358 | 0.308 | 0.169 | 0.088 | 0.344 | 11 |
| | T2 | 0.321 | 0.283 | 0.236 | 0.218 | 0.281 | 0.325 | 0.256 | 0.137 | 0.088 | 0.392 | 8 |
| | T3 | 0.321 | 0.278 | 0.275 | 0.257 | 0.273 | 0.299 | 0.246 | 0.128 | 0.086 | 0.403 | 7 |
| | T4 | 0.295 | 0.230 | 0.270 | 0.201 | 0.180 | 0.232 | 0.237 | 0.178 | 0.040 | 0.182 | 12 |
| | T5 | 0.334 | 0.266 | 0.258 | 0.246 | 0.252 | 0.322 | 0.231 | 0.137 | 0.087 | 0.388 | 9 |
| | T6 | 0.309 | 0.295 | 0.245 | 0.238 | 0.328 | 0.274 | 0.277 | 0.124 | 0.090 | 0.421 | 6 |
| | T7 | 0.286 | 0.306 | 0.295 | 0.311 | 0.202 | 0.250 | 0.224 | 0.143 | 0.082 | 0.364 | 10 |
| | T8 | 0.281 | 0.292 | 0.348 | 0.341 | 0.211 | 0.281 | 0.240 | 0.126 | 0.101 | 0.445 | 4 |
| | T9 | 0.266 | 0.313 | 0.280 | 0.294 | 0.441 | 0.282 | 0.291 | 0.092 | 0.137 | 0.599 | 2 |
| | T10 | 0.257 | 0.328 | 0.329 | 0.351 | 0.431 | 0.369 | 0.388 | 0.051 | 0.173 | 0.770 | 1 |
| | T11 | 0.232 | 0.318 | 0.271 | 0.285 | 0.302 | 0.231 | 0.349 | 0.120 | 0.094 | 0.438 | 5 |
| | T12 | 0.224 | 0.285 | 0.395 | 0.445 | 0.207 | 0.185 | 0.358 | 0.134 | 0.141 | 0.513 | 3 |
| | $A^+$ | 0.334 | 0.328 | 0.395 | 0.445 | 0.441 | 0.369 | 0.388 | | | | |
| | $A^-$ | 0.224 | 0.230 | 0.210 | 0.164 | 0.180 | 0.185 | 0.224 | | | | |
| | R | −0.650 * | 0.748 ** | 0.636 * | 0.762ns | 0.720 ** | 0.000 ns | 0.643 * | | | | |
| Second season | CK | 0.327 | 0.230 | 0.277 | 0.223 | 0.178 | 0.377 | 0.264 | 0.118 | 0.080 | 0.404 | 10 |
| | T2 | 0.330 | 0.296 | 0.270 | 0.286 | 0.186 | 0.360 | 0.263 | 0.103 | 0.083 | 0.446 | 8 |
| | T3 | 0.313 | 0.305 | 0.219 | 0.240 | 0.253 | 0.334 | 0.312 | 0.089 | 0.077 | 0.464 | 7 |
| | T4 | 0.307 | 0.277 | 0.270 | 0.261 | 0.230 | 0.246 | 0.315 | 0.105 | 0.057 | 0.353 | 12 |
| | T5 | 0.300 | 0.301 | 0.210 | 0.217 | 0.296 | 0.287 | 0.357 | 0.090 | 0.080 | 0.470 | 6 |
| | T6 | 0.264 | 0.307 | 0.257 | 0.296 | 0.330 | 0.303 | 0.279 | 0.065 | 0.088 | 0.575 | 3 |
| | T7 | 0.287 | 0.285 | 0.297 | 0.287 | 0.349 | 0.291 | 0.270 | 0.063 | 0.093 | 0.598 | 2 |
| | T8 | 0.254 | 0.281 | 0.329 | 0.312 | 0.239 | 0.256 | 0.294 | 0.096 | 0.068 | 0.414 | 9 |
| | T9 | 0.296 | 0.316 | 0.307 | 0.338 | 0.409 | 0.239 | 0.282 | 0.068 | 0.121 | 0.641 | 1 |
| | T10 | 0.307 | 0.292 | 0.319 | 0.321 | 0.323 | 0.252 | 0.286 | 0.072 | 0.093 | 0.563 | 4 |
| | T11 | 0.242 | 0.303 | 0.340 | 0.339 | 0.315 | 0.245 | 0.235 | 0.087 | 0.091 | 0.510 | 5 |
| | T12 | 0.211 | 0.260 | 0.333 | 0.312 | 0.266 | 0.228 | 0.289 | 0.103 | 0.069 | 0.401 | 11 |
| | $A^+$ | 0.330 | 0.316 | 0.340 | 0.339 | 0.409 | 0.377 | 0.357 | | | | |
| | $A^-$ | 0.211 | 0.230 | 0.210 | 0.217 | 0.178 | 0.228 | 0.235 | | | | |
| | R | −0.189 ns | 0.734 ** | −0.021 ns | 0.399 ns | 0.881 ** | −0.042 ns | −0.308 ns | | | | |

Notes: SW, TSS, OA, SAR, VC, Lyc and Fn indicated tomato single weight, total soluble solids, organic acids, solid–acid content ratio, vitamin C content, lycopene content and fruit firmness, respectively. * means statistically significant with $p < 0.05$; ** notes statistically extreme significance with $p < 0.01$; ns represents statistically insignificant with $p > 0.05$.

The heat map of net profit percentage change compared to CK under different treatments is shown in Figure 4. Yield dominated the comprehensive benefits when the market price sensitivity was low (R0~0.6), and the highest profit ratios found in T5 and T6 were 11% and 19% in the first and second seasons, respectively. Sufficient irrigation and mild nitrogen promised a relative high yield. As the sensitivity of price improved (R0.8~1), quality became more important than yield, and severe deficit treatment with high fruit quality reached the highest profit. The profit ratio of T10 reached 31% in the first season and 33% in the second season. Similar results were also found in grapes [50].

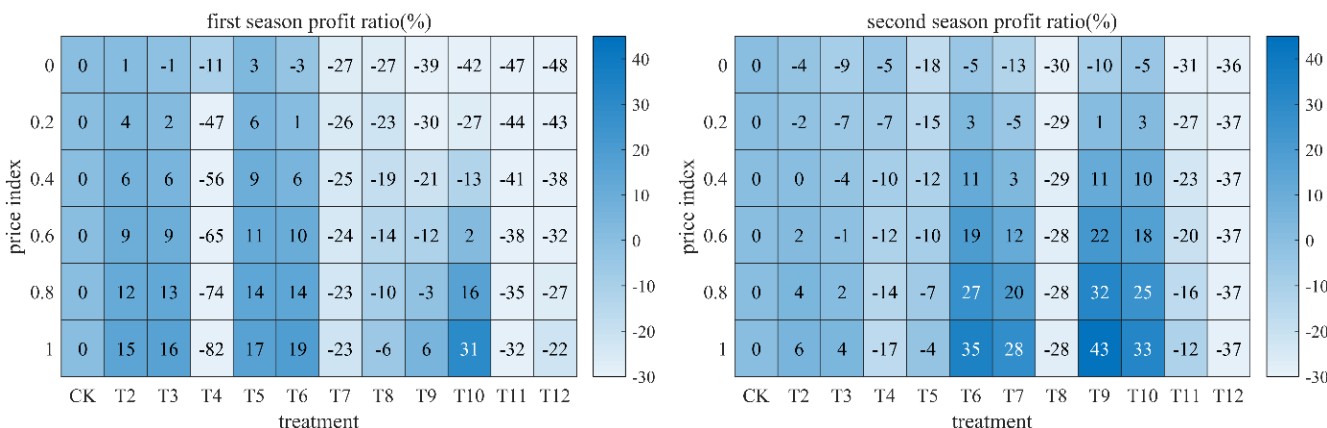

**Figure 4.** The tomato profit ratio in the first and second seasons of 2019.

## 4. Discussion

The results show that water and nitrogen application deficit decreased plant height, stem diameter, leaf area index and leaf chlorophyll content (3.1), and this is in agreement with previous studies [24,51]. Nitrogen, as an essential element in synthetic amino acids, protein and chlorophyll, could affect LAI and chlorophyll more seriously than water [27]. Since T12 (W1/3N0) always showed the lowest values in all treatments, the coupling effect of water and nitrogen was more obvious than either of them, which confirmed that under certain circumstances, nitrogen could cripple the inhibition caused by water deficit, and vice versa [40,42].

Evapotranspiration and yield decreased significantly with the decline in irrigation and nitrogen application (Section 3.2). Irrigation deficit could directly reduce root zone soil moisture and plant water use, and nitrogen stress decreased ET through restraining plant leaf development [34,52]. Water deficit decreased yield more seriously than nitrogen (Section 3.2), and the yield in the first season was lower than that of the second season, in line with the results in south Spain [53]. In the current study, the different ET and yield between the two seasons was due to the significant variances in meteorological parameters (Figure 1). The lower temperature and solar radiation in the second season reduced potential plant ET and led to diminished ET (Figure 1 and Table 2), photosynthesis and other meteorological processes, which finally caused a lower yield [54]. Water supply improved NUE, and nitrogen supply enhanced WUE (Section 3.2), which were also observed in previous water–nitrogen deficit experiments [55,56]. On the one hand, as a necessary mineral solvent, water directly participates in nitrogen uptake of plant root and restricts nitrogen use; on the other, nitrogen fertilization can promote plant root biomass, improve its absorptive capacity and enhance water use efficiency [28,57]. In addition to nitrogen, other fertilizers also play an important role in plant growth [58]. As soil–plant–air is a continuous system, the interactions between soil characteristic and fertigation still demand research [59].

Fruit quality parameters, e.g., TSS, SAR and VC, increased with the decline in water supply (Section 3.3). Reduced irrigation decreased water content in tomato fruit, which formed a concentration effect [7]. VC synthesis and the conversion of acid to sugar can be improved by more sugar and less water in fruit [52]. Regarding nitrogen, fruit quality increased first and then decreased with the deficit of nitrogen fertilizer (Section 3.3). Previous studies reported that nitrogen fertilizer provided essential biochemical material and enhanced tomato fruit qualities [7,25,31], while excess nitrogen fertilizer application decreased lycopene and VC content in fruit. The sufficient nitrogen level in this study probably exceeded the actual nitrogen demand according to conventional nitrogen application strategy, and thus, moderate nitrogen deficit treatments had a higher TSS and VC content compared with full nitrogen fertilization.

Although both water and nitrogen affected yield and fruit quality significantly, the integrated relationship between them only existed within a certain threshold (Section 3.4).

The stress adaptive capacity of plants was limited by physiological metabolic process, and water–nitrogen interaction mostly existed in the moderate deficit section. Once the deficit lever of a factor exceeded the threshold, the other influencing factor could probably no longer cooperate with or antagonize the deficit factor, and then, they only would affect plants separately [60–62]. This might explain the non-significance of the interaction effect of water and nitrogen in this study and many earlier studies [34,41,42]. For deeper research, a molecular biology study is needed to determine the variances in hormone signals, key enzyme activities, active genes in a plant when suffering deficit and the inner mechanism of fertigation application. TSS achieved dominance in comprehensive quality evaluation with a high weight and a large fluctuation of water–nitrogen deficit (Section 3.5), which could be the main proxy to judge fruit quality when measurements are limited. In economic analysis, the prices only referred to the local market in Shandong province, China, where the water and labor cost was relatively low and fertilizer cost was relatively high, and the final water and nitrogen application decision may not suitable to other places where the resources prices differ too much. Specific analysis was needed for localization.

## 5. Conclusions

Deficit irrigation and nitrogen application restrained tomato growth, including plant height, stem diameter and LAI, slightly. Tomato yield and partial quality indicators (SW and OA) decreased with the decline in water and nitrogen. For TSS and VC, water deficit had a promoting effect, while nitrogen deficiency showed an inhibitory impact. Water and nitrogen deficit positively impacted SAR. Water showed greater influence on tomatoes compared with nitrogen, and their integrated relationship was exhibited within the mild deficit threshold of N1~N1/3 and W1~W2/3.

Based on water and nitrogen use efficiency, severe water and nitrogen deficit (W1/3N1/3) was the best scheme; in terms of high fruit quality and net profit, severe water deficit and mild nitrogen deficiency (W1/3N2/3) was optimal. In summary, the 1/3 full irrigation and mild nitrogen deficit (N2/3, N1/3) treatment produced higher profits and is recommended to tomato cultivation industry.

**Author Contributions:** Conceptualization, X.J. and N.C.; methodology, Y.Q.; software, L.Z. and S.J.; validation, Y.W., Y.Z. and S.J.; formal analysis, M.F. and Y.Z.; investigation, Y.Q.; resources, X.J. and N.C.; data curation, X.J. and M.F.; writing—original draft preparation, M.F., Y.Q. and X.J.; writing—review and editing, M.F.; visualization, M.F.; supervision, Y.W. and N.C.; project administration, X.J., N.C. and L.Z.; funding acquisition, N.C. and Y.W. All authors have read and agreed to the published version of the manuscript.

**Funding:** This study was financially supported by the Key Research and Development Program of Beijing (Z20111000800000), the National Natural Science Foundation of China (51709203, 51922072, 51779161, 51009101), the Science and Technology Projects of Sichuan (22ZDYF0145, 22QYCX0069, 22QYCX0073, 22QYCX0115) and the Fundamental Research Funds for the Central Universities (2019CDLZ-10, 2020CDDZ-19).

**Institutional Review Board Statement:** Not applicable.

**Informed Consent Statement:** Not applicable.

**Data Availability Statement:** The data presented in this study are available on request from the corresponding author.

**Conflicts of Interest:** The authors declare no conflict of interest.

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
