# Peer review of "Proper Deficit Nitrogen Application and Irrigation of Tomato Can Obtain a Higher Fruit Quality and Improve Cultivation Profit"

_agronomy, doi:10.3390/agronomy12102578_

Round 1
Reviewer 1 Report
The manuscript is well-prepared, the experimental design is acceptable, the result analysis and discussion are reasonable. However, some minor issues underlined below need to be taken under consideration before publication, as follows:
English grammar and spelling should be checked
In Fig. 2 you mention first and second season, but in the text, lines 227..... you only mentioned second season.
Fig. 3 and 4 should be enlarged.
Line 360 should be amino acids.
Line 414 should be treatments not programs.
Reviewer 2 Report
Authors.
Indicate the affiliation of each author2. Material and methods
In general, the mathematical aspects of the material and methods are well described. However, the technical-agronomic part of the material and methods is poorly described. The material and methods must allow the test to be replicated. It is not possible to reproduce the experiment exactly with the information that appears in the paper.2.1 Experimental site
Different levels of nitrogen fertilization are evaluated in the article. For this it is necessary to know the previous levels of nitrogen in the soil. Could you provide the levels of mineral N and organic N in the starting soil?Lines 96, 97. What methodology has been used to determine the field water capacity and the wilting coefficient? How many samples have been taken for it?
2.2 Experimental design
What has been the planting framework of the crop? What is the framework of irrigation and flow of the emitters used? What was the end date of the growing cycles? Lines 110, 111. What is the flap material?Line 113. Does the local management include the application of organic matter in the soil before cultivation? If so, describe characteristics and amount provided.
Lines 116,117. If the number of irrigation and fertilization events is the same in all the treatments of the same season, how is the irrigation and fertilization amount different, reducing the irrigation time or using drippers with different flow rates?
Lines 118, 119 and Table1. If in the branch of eatch plot was separately installed with a water meter and a venturi fertilizing tank was used to record the irrigation and fertilizatión amount, Why do the amounts shown in table 1 of irrigation and nitrogen amount in the reduced treatments not coincide with 2/3 or 1/3? In fact, in the second season it is called 2/3 at 80% of the fertilizer and 1/3 at 60%.
2.3 Measurements
2.3.2 Evapotranspiration
Line 145. Drainage cannot be ignored because the irrigation method is drip irrigation. It can be ignored if irrigation management is done in which the field water capacity is not exceeded. This sentence needs to be corrected.
2.3.3 Plant growth
Line 148. What criteria has been followed to choose the point of measurement of the diameter of the stem? A measurement interval of 7-10 days is inaccurate. If the interval has not been constant, it is better to indicate the number of measurements in each season.
2.3.4 Yield, water and nitrogen use efficiency
Line 157. Must be corrected: (Y, t·hm-1). Replace with: (Y, t ha-1)
Equations 4, 5 and 6. In the same way that the efficiency in the use of water is differentiated from the efficiency in the use of irrigation water, the efficiency in the use of nitrogen should be differentiated from the efficiency in the use of fertilization nitrogen. For this it is necessary to provide the initial and final levels of nitrogen in the soil of each treatment.Line 161. Must be corrected: (Y, t·ha-2). Replace with: (Y, t ha-1)
2.5 Economic analysis
Line 200. Market sale price of tomato in 2019 is 10 $/kg?. It seems very tall. What is the reference of said market?
Line 208. Market sale price of N in 2019 is 15 $/kg?. It seems excessively high. What is the reference of said market? The international price of urea in 2019 remained below $0.25/kg and the nitrogen richness of urea is 46%.
Line 211-213. Although the rest of the cultivation costs are the same in the different treatments, their value must be indicated, or at least their sum.
3. Results
Figure 2. It is indicated that the graphs shown correspond to the two seasons, but in the figure only those of one appear. What season is the one shown? Why does the data of the other not appear?
Better to use in the figure the term prune than trim
3.2 Effect of deficit irrigation and nitrogen on evapotranspiration, yield, water and nitrogen use efficiency
Table 2. Substitute WUEi for WUEI to homogenize with line 273
It is very surprising that the production in the 0N treatments is statistically equal to the production in the 100% N treatments. This can only be the case if the soil is rich in nitrogen. Soil N levels, both initial and final, should be shown.
5. Conclusions
The first conclusion states:"Water and nitrogen deficit restrained tomato growth including plant height, stem diameter and LAI slightly." With the information that appears in the article, it can be affirmed that there has been a water deficit. But, can it be affirmed that there has been a deficit of N? Could it be that the levels in the soil have been so high that there has been no deficit, regardless of the contributions that have been made with fertigation?
Reviewer 3 Report
Dear author(s),
there are some inspiring insights thorough the manuscript and I tend to support its publication. Only few and quick changes are recommended to improve its communication:
Title:
1/ too long, too bold claim
Abstract:
2/ better follow the established schema of writing academic Abstract: A/ introduction (urgency and significance of the research hypothesis); B/ principles of the methods used + key results; C/ conclusions (commercial and environmental impacts)
3/ reduce the use of abbreviations and technical terms, please understand that the purpose of the Abstract is to explain to all readers (including those from other disciplines) what the paper is about
4/ there is no reason to go into detail and present the results obtained under specific reaction conditions, rather provide a synthesis of the results obtained
5/ better highlight the urgency and significance of your work, clearly indicate who (and how) will benefit from these findings (explain the importance)
Introduction:
6/ remove all clusters of references to avoid reference overkill (always use only 1 reference to support 1 claim)
7/ it should be better explained that N comes in different chemical forms, be more explanatory and deeper review the latest trends in nutrients recycling, refer to paper "Economic considerations on nutrient utilization in wastewater management"
8/ go straight to the point and more in depth, write more technically (always provide corresponding numbers), significantly condensate all the text by reducing ballast phrases and cliché
9/ it should be better explained that interactions with other nutrients are also important, refer to paper "Economic impacts of soil fertility degradation by traces of iron from drinking water treatment"
10/ make sure that this chapter fully introduces any reader into to the topic, explain all the terms, units, abbreviations and the whole context that is necessary for anyone (including experts from other disciplines) to understand the following chapters
11/ deeper review the interactions with soil/substrate, refer to paper "Techno-economic analysis reveals the untapped potential of wood biochar"
12/ the research hypothesis is not clearly stated, clearly justify the urgency and importance of its investigation, clearly identify those who will benefit from the findings
Materials and Methods:
13/ the method must be presented in such a way that it can be reproduced anytime, by anyone, anywhere (do not create obstacles like referring to specific location, season or time period etc.)
14/ each material/reactant and apparatus used needs to be presented in detail (serial number, setup, process parameters, manufacturer, country of origin, purity etc.)
15/ provide cost breakdown or at least some simplified financial analysis if you are about to argue that this concept is realistic
Results:
16/ each Tab. and Fig. should be provided with caption that describes A/ what can be seen and B/ how is this relevant to the research hypothesis
17/ avoid data overkill, present only the most most industrially important results
Discussion:
18/ show more self-criticism to your work (can all the methods and results be fully trusted? what are the weaknesses of the methods used? where do the main measurement inaccuracies arise? what are the limitations from a commercial point of view? are the lessons learned transferable to other fields?)
19/ propose some improvements and direction for future research, refer to paper "Economic impacts of soil fertility degradation by traces of iron from drinking water treatment"
20/ compare your results in more depth with the existing literature, identify the main deviations and try to explain the mechanisms by which they may have been caused
Conclusions:
21/ reveal the main driving mechanisms of your results, provide deeper synthesis and reveal some more original/significant findings
22/ do not repeat your methods and results again and again, please understand that the Conclusion chapter is not a summary of your work, present only original and industrially significant revelations that have the potential to expand the horizon of human knowledge (higher level of generalization is mandatory)
23/ clearly indicate whether the research hypotheses tends to be confirmed or not
Round 2
Reviewer 2 Report
Line 111. What methodology has been used to determine the available nitrogen? Can the value of available nitrogen be considered high or low? It would be advisable to interpret the result of the analysis
Lines 127, 135, 138. What is the number of plants in each plot? Define in detail the separation between lines of plants and between plants within the line. Define in detail the separation between lines of drippers and between drippers within the line.
Line 145. Replace g·m2 with g·m-2
Table 2. How have NUE values ​​been obtained in the second season, in treatments 4, 8 and 12, if nitrogen has not been applied in these treatments? This needs to be clarified and corrected
